# Intermolecular interactions probed by rotational dynamics in gas-phase clusters

Chenxu Lu[1], Long Xu [2], Lianrong Zhou[1], Menghang Shi[1], Peifen Lu[1], Wenxue Li[1], Reinhard Dörner [3], Kang Lin [4] ✉ & Jian Wu [1,5,6] ✉

The rotational dynamics of a molecule is sensitive to neighboring atoms or molecules, which can be used to probe the intermolecular interactions in the gas phase. Here, we real-time track the laser-driven rotational dynamics of a single $N_2$ molecule affected by neighboring Ar atoms using coincident Coulomb explosion imaging. We find that the alignment trace of N-N axis decays fast and only persists for a few picoseconds when an Ar atom is nearby. We show that the decay rate depends on the rotational geometry of whether the Ar atom stays in or out of the rotational plane of the $N_2$ molecule. Additionally, the vibration of the van der Waals bond is found to be excited through coupling with the rotational N-N axis. The observations are well reproduced by solving the time-dependent Schrödinger equation after taking the interaction potential between the $N_2$ and Ar into consideration. Our results demonstrate that environmental effects on a molecular level can be probed by directly visualizing the rotational dynamics.

It has always been a great challenge to probe the intermolecular interactions in the gas phase due to the weak binding in atomic or molecular clusters[1,2]. High-resolution rotational or vibrational spectroscopy shows distinct fingerprints of the intermolecular interactions on the frequency domain[3,4]. Already two decades ago, a pioneering experiment has shown how many helium atoms are needed to show superfluidity. This was achieved by using the rotational and vibrational spectroscopy of a single carbonyl sulfide (OCS) molecule dissolved in helium clusters[5]. The solute of a single OCS molecule inside the helium solvent serves as a probe to sense the interaction with the surrounding atoms. Although the spectroscopy measurements provide a powerful tool for exploring the environmental effect, previous studies were mostly focused on the frequency domain while lacking the counterpart of dynamics information[6–8]. Rotational dynamics of molecules doped inside or on the surface of the helium nanodroplets has been addressed only recently[9–12]. By comparing isolated molecules with helium solute ones, these studies found the alignment traces of the in-droplet molecules turns out to be greatly broadened and followed by weak

oscillations of revivals[13,14]. From these observations, the authors concluded that a thin shell of helium atoms, resulting from the van der Waals (vdW) bonding between the embedded molecule and the closest neighboring helium atoms, corotates with the dopant molecule.

The complex dynamics of the helium droplets makes it difficult to pinpoint the most fundamental parts of the in-droplet dynamics. To obtain a clean picture of the environmental effect on the rotational dynamics, the time-dependent alignment factor of a molecule-atom complex of $C_2H_2$–He has been tracked for hundreds of picoseconds, and additional transition frequencies have been observed comparing to the isolated $C_2H_2$ in its Fourier transformed spectra[15]. It is numerically shown that the rotational dynamics of a molecule in a weakly bound molecule–atom complex will be severely affected[16,17]. Recently, investigations on the rotational dynamics of such floppy molecule–atom clusters have been fueled due to their unique properties under the interaction with neighboring environment[18–22].

In this work, we avoid the complexity added by the internal degrees of freedom of the droplet and instead study the influence of

¹State Key Laboratory of Precision Spectroscopy, East China Normal University, Shanghai, China. ²Department of Physics, Xiamen University, Xiamen, China. ³Institut für Kernphysik, Goethe-Universität Frankfurt am Main, Frankfurt am Main, Germany. ⁴School of Physics, Zhejiang Key Laboratory of Micro-Nano Quantum Chips and Quantum Control, Zhejiang University, Hangzhou, China. ⁵Chongqing Key Laboratory of Precision Optics, Chongqing Institute of East China Normal University, Chongqing, China. ⁶Collaborative Innovation Center of Extreme Optics, Shanxi University, Taiyuan, China. ✉e-mail: klin@zju.edu.cn; jwu@phy.ecnu.edu.cn

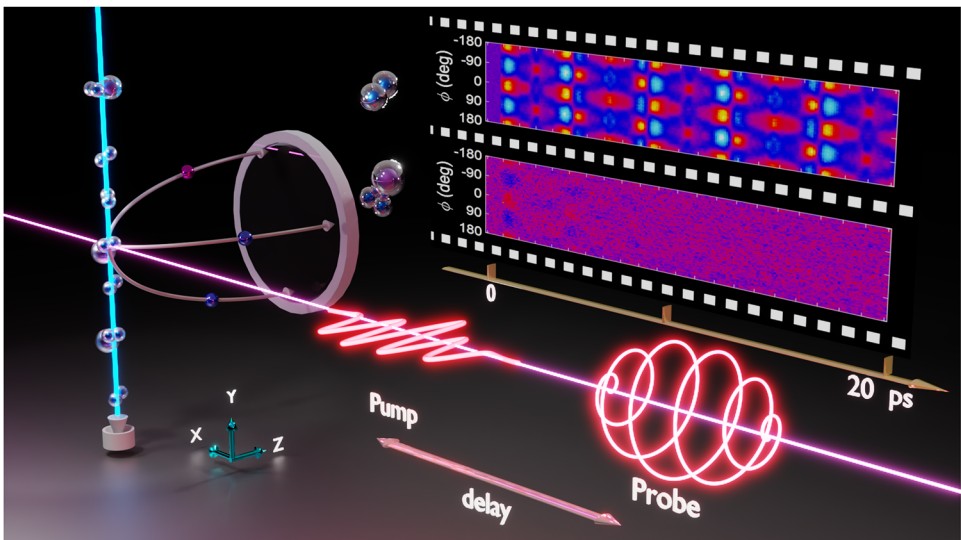

**Fig. 1 | Schematic diagram of the experiment.** A supersonic gas mixture of $N_2$ and Ar is impulsively excited by a Z-polarized pump pulse and the rotational dynamics of molecules is subsequently measured by a circularly polarized probe pulse. The inset on the top-right shows the measured time-resolved angular distributions of the N–N axis in isolated $N_2$ molecules (top) and $N_2$–Ar dimers (bottom).

the surrounding on rotation in the most minimal possible environment of a single atom. We report on a direct measurement of the rotational dynamics of a single $N_2$ molecule surrounded by no (isolated $N_2$) or one ($N_2$–Ar) Ar atom by coincident Coulomb explosion imaging (CEI)[23,24]. The rotational dynamics of the $N_2$ molecule serves as a probe to sense the interaction with the neighboring Ar atoms. We find the neighboring effect on the rotational dynamics is significant and differs from that of isolated ones. The alignment degree is greatly suppressed and decays fast when the $N_2$ molecule is surrounded by one Ar atom. Only the first few alignment peaks are observed, whereas no revival appears. This dephasing process is found to strongly depend on the rotational geometry of the $N_2$ molecule with respect to the Ar atom. When the Ar atom is in the rotational plane of the $N_2$ molecule, the alignment trace decays faster than the case when the Ar atom is out of the rotational plane. We also find that the stretching of the vdW bond is excited through coupling with the rotation of $N_2$ molecule.

## Results

Figure 1 shows the schematic diagram of the experimental setup. We only give a brief description here (see "Methods" for details). A non-ionizing and nonresonant linearly polarized pump pulse is focused onto a supersonic gas jet of $N_2$ and Ar mixture to trigger the rotational dynamics. The snapshots of the spatiotemporal evolution of the rotational wavepackets are taken via coincident CEI by using a much stronger time-delayed circularly polarized probe pulse. The three-dimensional momentum vectors of the fragment ions are measured using a reaction microscope of cold target recoil-ion momentum spectroscopy (COLTRIMS)[25,26]. Coulomb exploded fragments from isolated $N_2$ molecules and $N_2$–Ar dimers are discriminated using coincidence measurement.

The raw data are depicted as a "carpet" of the probability distribution, as shown in Fig. 1, where the horizontal axis is the time delay between the pump and probe pulses and the vertical axis is the azimuth angle $\phi$ of the N–N axis with respect to the Z axis in the polarization plane of the circularly polarized probe pulse (Y–Z plane). The pump pulse is linearly polarized along the Z axis ($\phi = 0°$, $\pm180°$). The yield is given by the color code. The top and bottom carpets in Fig. 1 show the time-resolved angular distributions of the N–N axis of the isolated $N_2$ molecules and that of the $N_2$–Ar dimers retrieved from the coincidently measured ($N^+$, $N^+$) and ($N^+$, $N^+$, $Ar^+$) channels, respectively. The time delay scans from −1 to 20 ps, covering more than two revival

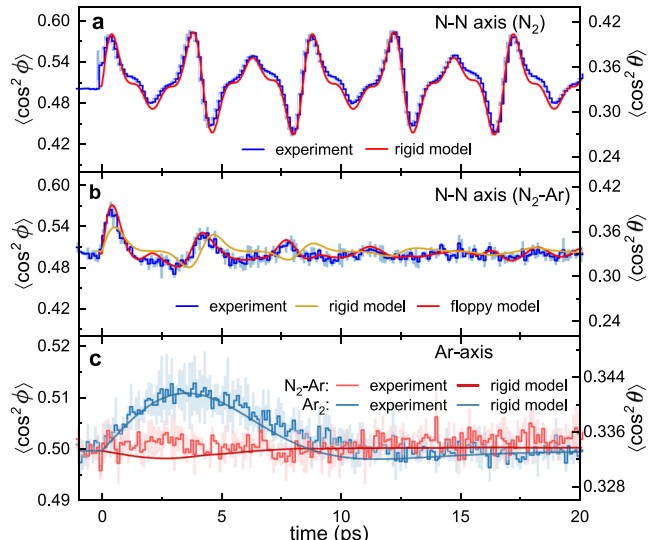

**Fig. 2 | Alignment traces of the N–N axis and Ar-axis for an isolated $N_2$ molecule, $Ar_2$ dimer, and $N_2$–Ar dimer.** Measured (horizontal step curves) and calculated (smooth curves) time-dependent alignment traces of N–N axis for **a** isolated $N_2$ molecules and **b** $N_2$–Ar dimers. **c** Measured (horizontal step curves) and calculated (smooth curves) time-dependent alignment traces of the Ar-axis of $N_2$–Ar and $Ar_2$ dimers. The shaded areas in (**a–c**) represent the standard deviations of three data sets.

periods of 8.39 ps for isolated $N_2$ molecules. Regular angular distribution as a function of the time delay can be clearly resolved for the case of isolated molecules, which is known as rotational revivals where alignment and anti-alignment appear periodically[27–29]. However, the situation changes dramatically for the case of when the $N_2$ molecule is surrounded by an Ar atom. Only angular focusing around the first few picoseconds can be discriminated, followed by an isotropic distribution. The difference indicates that the neighboring Ar atom hinders the free rotation of the N–N axis.

Figure 2a, b shows the one-dimensional alignment traces of the N–N axis of isolated $N_2$ molecules and that from $N_2$–Ar dimers, calculated from the top and bottom carpets in Fig. 1, respectively. The ensemble-averaged expectation value $<\cos^2\phi>$ is used to characterize

the alignment degree, where $<\cos^2\phi> = 0.5$ indicates an isotropic angular distribution, while the $<\cos^2\phi>$ larger or smaller than 0.5 corresponds to alignment or anti-alignment, respectively. Quantum mechanically, the linearly polarized pump pulse induces an umbrella time breathing between a prolate and oblate distribution of the N–N axis with respect to the pump axis. From this intricate three-dimensional quantum dynamics, our detection by a circularly polarized probe pulse and the inspection of the angle $\phi$ probes a motion that would classically correspond to a rotation in that plane, either clock or counterclockwise. We find that the alignment degree of the N–N axis in a $N_2$–Ar dimer is greatly suppressed and shows a fast decay as compared to that of the isolated $N_2$ molecules, confirming the hindered rotation by the neighboring Ar atom. On the other hand, Fig. 2c shows that the vdW axis between the $N_2$ molecule and the Ar atom, termed as Ar-axis hereafter, hardly responds to the kick. It is worth to mention that our observation window is long enough if the Ar-axis shows an immediate post-pulse alignment. As a reference, the

vdW axis of the $Ar_2$ dimer with a similar rotational constant, interaction strength and comparable polarizability anisotropy[22,30–32] shows a clear alignment peak at ~3.5 ps. The drastically different alignment responses for the N–N axis and the Ar-axis inside the $N_2$–Ar dimer from that for isolated $N_2$ molecule and $Ar_2$ dimer indicate that the rotational dynamics can serve as a sensitive probe of the molecule–atom interaction.

Intuitively, the laser-induced rotation of the N–N axis suffers smaller resistance when the Ar-axis is perpendicular to other than in the rotational plane, behaving like a copter. Our measurement indeed confirms the above assumption. Benefiting from the statistically super weak alignment response of the Ar-axis, we found that the difference between the in- and out-of-plane rotations survives after selecting the relative angle between Ar-axis and the probe plane. As shown in Fig. 3, for the out-of-plane geometry, the alignment degree is higher than that of the in-plane one. The decay of the in-plane geometry is also faster than that of the out-of-plane one. Intuitively, for the out-of-plane geometry, the rotation of the N–N axis does not change the original T-shape structure of the $N_2$–Ar dimer, whereas the Ar-axis is severely bended for the in-plane geometry that the potential barrier hinders the rotation in return. After being triggered by the alignment pulse, the N–N axis starts to rotate towards the laser polarization direction, which gives rise to a structural deformation of $N_2$–Ar if the torque can overcome the vdW potential barrier. However, the potential barrier between the T-shape and linear configuration of $N_2$–Ar is much higher than the rotational energy of the $N_2$ molecule[33]. Thus the structure of $N_2$–Ar is slightly distorted, resulting in a smaller alignment degree.

In the following, we Fourier transform the alignment traces in Fig. 2a, b, as shown in Fig. 4a. Three main peaks are observed in the power spectra. The central value of each spectral peak is assigned to the frequency of a $j$–$j + 2$ coherence, where $j$ is the rotational quantum number of $N_2$. For the isolated $N_2$ molecules, we assign the three observed peaks to 0–2, 1–3, and 2–4 coherences, respectively. When it comes to the $N_2$–Ar dimers, the three frequency peaks are red-shifted as compared to the case of isolated $N_2$ molecules, corresponding to narrower gaps between the energy levels induced by the neighboring Ar atom. Classically, such narrower energy gaps can be assumed as a

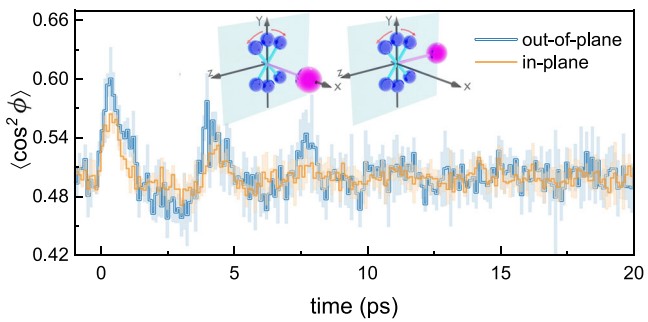

**Fig. 3 | Rotational geometry effects on alignment traces of the N–N axis.** Measured time-dependent alignment traces of the N–N axis for in- and out-of-plane rotational geometries. The shaded areas represent the standard deviations of three data sets. The insets schematically illustrate the out-of-plane (left) and in-plane (right) geometries, where the shaded plane represents the polarization plane of the probe pulse. The $N_2$ molecule and Ar atom are colored in blue and purple, respectively.

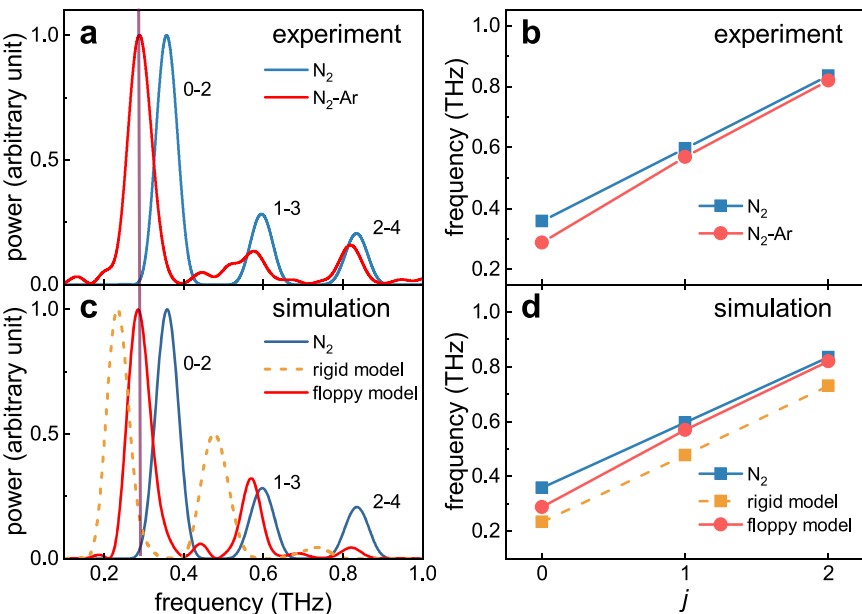

**Fig. 4 | Rotational spectra of the N–N axis. a, c** Power spectra of the measured and calculated alignment traces of the N–N axis for isolated $N_2$ molecules and $N_2$–Ar dimers. The central frequencies of peaks are labeled with $j$–$j + 2$ coherences. **b, d** Central frequencies of peaks in the corresponding power spectra versus $j$ for isolated $N_2$ molecules and $N_2$–Ar dimers. In simulations, the rigid model considers a rigid connection between $N_2$ and Ar, while the floppy model takes the intermolecular interaction into consideration.

**Table 1 | Moments of Inertia and Polarizabilities of the $N_2$–Ar dimer under T-shape configuration**

| Moments of inertia | Polarizability |
|---|---|
| $I_a$ = 225.54 | $\alpha_{aa}$ = 3.71 |
| $I_b$ = 8.47 | $\alpha_{bb}$ = 3.32 |
| $I_c$ = 234.02 | $\alpha_{cc}$ = 3.04 |

The orthogonal a-axis and b-axis represent the axes parallel to the N–N axis and Ar-axis in the molecular frame, respectively. The c-axis is perpendicular to the a-axis and b-axis. $I_i$ represents the moment of inertia (in amu Å$^2$) of the i-axis, while $\alpha_{ii}$ represents the corresponding polarizability (in Å$^3$) along this axis, i = a, b, c. Moments of inertia were computed using atomic masses and their Cartesian coordinates, N = (±0.55,0,0) and Ar = (0,3.7,0) in Å.

slower rotation in time domain. In addition, the shift in frequency varies with different $j$ coherences. As shown in Fig. 4b, the frequency difference between $N_2$ and $N_2$–Ar gradually decrease as the $j$ increases. It implies that the slower rotation imposed by the neighboring Ar atom in the $N_2$–Ar dimer depends on the rotational frequency of $N_2$ molecule. Higher rotational states behave more analogously to an isolated molecule[34]. An intuitive analog is to consider the rotating molecule as a gyroscope[35]. The faster it rotates, the harder it can be altered by the environment.

To understand the underlying physics of the rotational dynamics of the $N_2$ molecule under the influence from the neighboring Ar atom, we performed rigid and floppy model simulations separately. In the rigid model, the $N_2$–Ar dimer is treated as a rigid asymmetric-top molecule. On the other hand, the vdW bonding nature of the $N_2$–Ar dimer is represented by the molecule–atom interaction potential in the floppy model, where the dynamics of $N_2$-Ar dimer is described by the rotation of $N_2$, rotation of Ar-axis and intermolecular stretching (see "Methods"). Different from the case of high gas densities and temperatures where the collisions dominate the molecule–atom interactions[36,37], we only consider the interaction of a single $N_2$ molecule with one Ar atom at low temperatures when the vdW bond is formed.

Figure 2 shows the simulation results for both the rigid and floppy models. Although both results can repeat the decayed alignment traces, the rigid one shows alignment peaks mismatched from the measurement while the floppy model quantitatively agrees with the experiment. This indicates that the decayed alignment trace results from irregular energy spacing induced by the very unique asymmetric-top property of the $N_2$-Ar molecular structure, i.e., the moment of inertia of the Ar-axis differs drastically from the other two axes, as listed in Table 1. However, such asymmetric-top property cannot fully account for the observed rotational dynamics. Figure 4c, d shows the rotational spectra for the two models, which clearly illustrates that the rigid model fails to describe the influence of the neighboring Ar atom on the rotational energy of the N–N axis. Intuitively, the rigid model overestimates the red-shift because the Ar-axis is considered to completely corotate with the N–N axis which slows down its rotation in return. Only if the molecule–atom interaction potential rather than a rigid connection between $N_2$ and Ar is added to describe their interaction (see "Methods" for details of the floppy model), the rotational spectrum can be reproduced. The comparison between the two models unambiguously reveals that the rotational dynamics of $N_2$ molecule can serve as a sensitive probe of the neighboring environment. When it comes to the rotation of the Ar-axis, the rigid model considering the total polarizability shows a neglectable alignment response compared with the $Ar_2$ dimer. Thus, the vanishing alignment of the Ar-axis stems from the large discrepancy in moments of inertia, i.e., ~30 times smaller moment of inertia along the Ar-axis than that along the other axes. The laser pulse induces the alignment of the N–N axis through the fast rotation about the Ar-axis. However, the large moments of inertia of the other two axes lead to the neglectable change in the angular velocities of rotation about the other two axes.

The kick intensity used here is very low, which might also be the reason that leads to neglectable overall rotation. However, since the Ar-axes in $N_2$-Ar and Ar-Ar have similar interaction strengths and their anisotropies are comparable, as well as the kick intensities are the same, it is the large discrepancy of the three molecular axes of $N_2$-Ar compared with linear diatomic Ar-Ar that mainly leads to the weak alignment of the Ar-axis rather than the kick intensity. On the other hand, when increasing the kick intensity, the rotational dissociation happens[16,17].

## Discussion

For isolated diatomic molecules, the large energy gap ensures decoupling between rotation and vibration. As a result, the vibration can be treated individually under an identical radial potential consisting of orthogonal vibration states. However, this assumption does not hold any more when it comes to the $N_2$–Ar dimer since the molecule–atom interaction relies on both their relative distance and angle, i.e., the equilibrium distance between the $N_2$ molecule and Ar atom changes with the relative angle between the Ar-axis and the N–N axis. By assuming the repulsive potential of the two-body fragment channel $N_2$-Ar$^{2+}$ → $N_2^+$ + Ar$^+$ as a simple Coulomb potential of $1/r$, the intermolecular distance $r$ as a function of time can be reconstructed. Figure 5a shows the expectation value $<r>$ as a function of time, which indicates strong vibrational excitation. Although only the polarizability of $N_2$ is considered in our floppy model, the simulation result well reproduces the intermolecular stretching and its frequency components, as shown in Fig. 5a, b. This indicates that the intermolecular stretching is excited through the coupling with the rotation of the $N_2$ molecule. As marked by the blue arrows in Fig. 5c, states with energy gaps of 15.8 and 20.1 cm$^{-1}$ are assigned to the frequency components $v_1$ and $v_2$, respectively. The frequency $v_2$ originating from the beating between $n = 0$ and 1 dominates the intermolecular stretching, where $n$ is the approximate stretch quantum number representing the number of nodes in the stretch coordinate $r$[34]. The frequency $v_1$ originates from the identical approximate stretch quantum number $n = 0$ of similar but different radial distributions (see Supplementary information) as $n$ is not rigorous, which results in a relatively smaller amplitude as compared to $v_2$. The superposition of these states results in the observed forth-back stretching in Fig. 5a.

Figure 5c displays the transition diagram of the ro-vibrational spectra obtained from the space-fixed expression[38]. By keeping the low kick intensity to avoid rotational dissociation[16,17], we ensure that the observed dynamics stays in bound states of $N_2$-Ar. Since the angular part is expanded onto the coupled basis in the floppy model, approximate quantum numbers $(n, j, L)$ are employed to label the eigenstates corresponding to the intermolecular stretching, $N_2$ rotation and vdW rotation, where $j$ and $L$ are the approximate rotational quantum numbers of $N_2$ and Ar-axis. Only the parity $p = (-1)^{j+L+J}$, the total angular momentum $J$ and its projection $N$ are rigorous. As numerous states are involved, for illustration, only several states with even parity and $J = 4$ are displayed. In the space-fixed expression, the quantum numbers $j$ and $L$ are not rigorous which means that an eigenstate of $N_2$-Ar is a superposition of different rotational states of the N–N axis and Ar-axis (see "Methods"). For simplicity, the approximate quantum number $j$ and $L$ are assigned according to their most populated components (see Supplementary information). The approximate quantum number is instructive as it builds up a connection between isolated $N_2$ and $N_2$–Ar dimer, and reveals behaviors of the N–N axis with different angular momentum under the interaction of Ar. Besides the space-fixed expression, a molecular frame knowledge about the internal bending of $N_2$–Ar dimer can be obtained under the body-fixed expression[32,34,39]. For the rotation of $N_2$, states with energy gaps of 9.27, 17.85, and 26.04 cm$^{-1}$ have been marked by red arrows in Fig. 5c, which correspond to the observed rotational frequencies in Fig. 4. Collaborating with the approximate quantum number $j$ of $N_2$-Ar, these frequencies can relate to the ones of the $N_2$ monomer

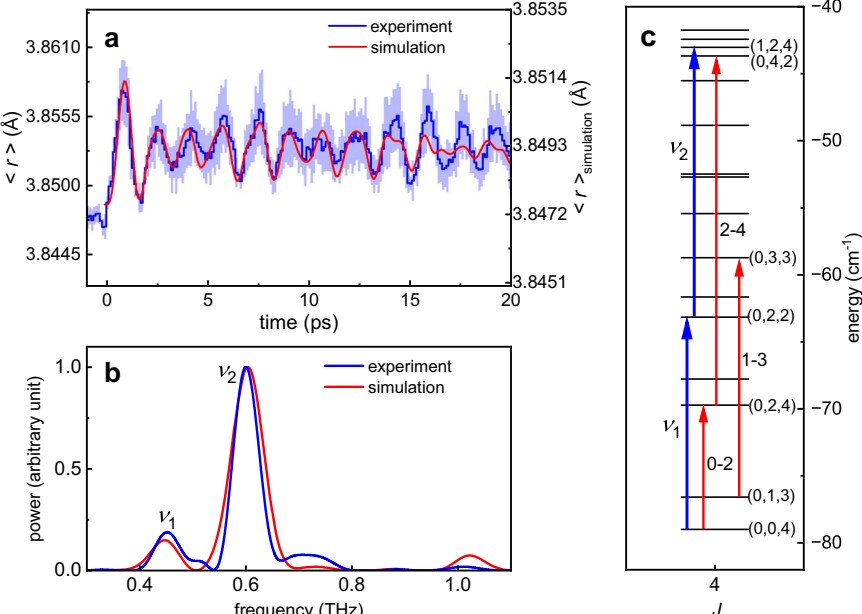

**Fig. 5 | Intermolecular stretching of the $N_2$-Ar dimer. a** Reconstructed (blue curve) and calculated (red curve) time-dependent intermolecular stretching of the $N_2$–Ar dimer. The shaded areas represent the standard deviations of three data sets. **b** Power spectra of the reconstructed (blue curve) and calculated (red curve) intermolecular stretching. Two central frequencies of peaks are marked by $v_1$ and $v_2$. **c** The energy levels, with the even parity $p = (-1)^{j+L+j}$ and $J = 4$, are grouped by the approximate quantum numbers $(n, j, L)$. The energy gaps related to the frequencies of rotation of $N_2$ and intermolecular stretching are marked by red and blue arrows, respectively. For comparison with gas-phase molecules, $j–j + 2$ coherences are used to label states contributing to the rotation of $N_2$ in $N_2$–Ar. The states related to the intermolecular stretching are labeled with $v_1$ and $v_2$.

with 0–2, 1–3, and 2–4 coherences. Thus, the red-shift of rotational frequencies results from smaller energy gaps for states of the $N_2$–Ar than the $N_2$ monomer.

In conclusion, we use a single $N_2$ molecule as a probe to sense its interaction with the surrounding Ar atom by observing its rotational dynamics. Due to the interaction with a neighboring Ar atom, the alignment degree of $N_2$ molecule is greatly suppressed as compared to that of the isolated one. We also show the rotational geometry effect of the neighboring atom on the rotational dynamics. Our result directly shows the field-free evolution of the distorted rotational wavepackets as a function of time on its intrinsic timescale. Furthermore, we demonstrate the coupling between $N_2$ rotation and the intermolecular stretching, through which the intermolecular vibration can be excited. Our work could open the opportunity to use rotational manipulations, including the unidirectionally rotating molecules[40–43] and rotational echoes[44–48], as a tool to probe the more complex molecule–atom dynamics.

## Methods

### Experimental setup

The experiment was performed with ultrashort femtosecond laser pulses (25 fs, 790 nm, 10 kHz) produced from a Ti:sapphire multipass amplifier laser system. The laser beam was separated into the pump and probe arms via a beam splitter with an intensity ratio of 7:3, after which the pump and probe pulses were recombined in a Mach–Zehnder type interferometer configuration. The intensities of them were adjustable using two neutral density filters and a quarter wave plate is placed in the arm of the probe pulse to make it circularly polarized. The probe pulse additionally passed through a telescope which increased its diameter by a factor of 1.5. As depicted in Fig. 1, the pump pulse was linearly polarized along the $Z$ axis and the probe pulse was circularly polarized. The time delay between the pump and probe pulse was controlled by a motorized delay stage which scanned from −1 to 20 ps with a step size of 100 fs. The negative time delay, where the probe pulse is prior to the alignment pump pulse, was employed to

eliminate the signal bias induced by the imperfect circularity of the probe pulse and calibrate reconstructed internuclear distances with the simulation results. The peak intensities of the pump and probe pulse were estimated to be $7 \times 10^{12}$ and $6 \times 10^{14}$ W/cm$^2$, respectively. Clusters are produced via a supersonic expansion of a gas mixture with $N_2$:Ar ~1:1 through a 30-μm-diameter nozzle at a driving pressure of 4 bar. The initial temperatures of $N_2$ and $N_2$–Ar are estimated to be 11 K and 7 K, respectively, by approximation to their translational temperatures of $T_{trans} = \Delta p^2/[4\ln(4)k_B m]$[49], where $k_B$ is the Boltzmann constant, and $\Delta p$ and $m$ are the full width at half maximum (FWHM) of the momentum distribution in the jet direction ($Y$ axis in our case) and mass of the singly ionized molecules of $N_2^+$ and $N_2$–Ar$^+$. In our experiment, we measure $\Delta p$ ~3.2 a.u. and $\Delta p$ ~4.0 a.u. for $N_2^+$ and $N_2$–Ar$^+$ ions. We expect the measured translation temperature is similar to the rotational and vibrational temperature of molecules in the supersonic gas jet[50–52]. The excellent agreement with the experiment can be achieved in the simulation when using the temperatures of 9 K for isolated $N_2$ and 7 K for $N_2$–Ar molecules. The pump and probe laser pulses were focused onto the supersonic beam by a concave mirror with a focal length of 75 mm inside the ultrahigh-vacuum chamber of a cold target recoil-ion momentum spectroscopy (COLTRIMS). Coulomb explosion channels of $(N^+, N^+)$ and $(N^+, N^+, Ar^+)$ were selected to retrieve the time-resolved spatial distribution of $N_2$. The orientation of N–N axis was retrieved from momentum difference between two $N^+$ fragments from the above two channels accordingly. The Ar-axis of a $N_2$–Ar and Ar$_2$ dimer were obtained from the two-body explosion channel $(N_2^+, Ar^+)$ and $(Ar^+, Ar^+)$. For the analysis of the in-plane geometry, the events with three fragments confined to [−40°, 40°] with respect to the $Y$–$Z$ plane were selected, while for the out-of-plane geometry, the $N^+$ and $Ar^+$ ions were confined to [-40°, 40°] with respect to the $Y$–$Z$ plane and the $X$ axis, respectively. The ionic fragments created by strong-field ionization were guided by a weak homogeneous electric field to a time- and position-sensitive microchannel plates (MCP) detector at the end of the spectrometer. The Coulomb explosion channels of the multiply ionized clusters were

identified based on the momentum conservation of the measured ionic fragments.

## Rigid model

In the rigid model, the $N_2$–Ar molecules are modeled as rigid asymmetric tops with anisotropic polarizabilities and moments of inertia listed in Table 1[53,54]. The rotation of a rigid body was described by the Hamiltonian $\mathscr{H}_{rigid} = \mathscr{H}_0 + \mathscr{H}_{int}$ including the field-free Hamiltonian $\mathscr{H}_0 = L_a^2/2I_a + L_b^2/2I_b + L_c^2/2I_c$ and the interaction term of the molecule with the external field $\mathscr{H}_{int} = -\frac{1}{2}\sum_{ij}\alpha_{ij}E_iE_j$. Here $L_i$ is the angular momentum about $i$-axis with its moment of inertia $I_i$, while $E_i$ and $\alpha_{ij}$ represent the components of the field vector and polarizability tensor, $i = a, b, c$. The orthogonal $a$-axis and $b$-axis represent the axes parallel to the N–N axis and Ar-axis in the molecular frame, respectively. The $c$-axis is perpendicular to the $a$-axis and $b$-axis. The rotational dynamics of diatomic $N_2$ and $Ar_2$ were conducted under their own anisotropic polarizabilities and rotational constants[22,55].

## Floppy model

The floppy model is conducted under the space-fixed scheme, where the rotation of $N_2$ and the translation motion are included, and the molecule–atom interaction is represented by an accurate potential surface. Due to the complex formation, the angular part and the radical part of the translation motion are reduced to the rotation and intermolecular stretching of the vdW axis, respectively. With the rigid rotor approximation, the vibration of $N_2$ molecule is assumed to be neglected. Thus, the floppy model includes five degrees of freedom including two rotational degrees of $N_2$, two rotational degrees and one vibrational degree of vdW axis, and the Hamiltonian in the space-fixed scheme takes the form of

$$\mathscr{H} = \mathscr{H}_j + \mathscr{H}_L + \mathscr{H}_s + V(r,\theta_r) + W(\theta,t) \tag{1}$$

where $\mathscr{H}_j = \frac{Bj^2}{\hbar^2}$ is the rotational Hamiltonian of the $N_2$ molecule, $B = 1.99\ cm^{-1}$ is the rotational constant of the molecule, **j** is the angular momentum of the molecule; $\mathscr{H}_L = \frac{L^2}{2\mu r^2}$ is the Hamiltonian describing the rotation of the axis about the center of mass of the Ar atom and $N_2$ molecule, **L** is its angular momentum, $\mu$ is the molecule–atom reduced mass; $\mathscr{H}_s = -\frac{\hbar^2}{2\mu r}\frac{\partial^2}{\partial r^2}r$ is the Hamiltonian describing intermolecular stretching, $r$ is the intermolecular distance between the molecule and the atom; $V(r,\theta_r)$ is the potential energy surface for the molecule–atom interaction[31], $\theta_r$ is the angle between the N–N axis and Ar-axis, $W(\theta,t) = -\frac{E(t)^2}{4}(\triangle\alpha\cos^2\theta + \alpha_\perp)$ is the time-dependent molecule-laser potential, $E(t)$ is the field amplitude of the pump pulse, $\theta$ is the angle between the N–N axis and laser polarization, and $\triangle\alpha = \alpha_\parallel - \alpha_\perp$ is the polarizability anisotropy, with $\alpha_\parallel = 2.38\ \text{Å}^3$ and $\alpha_\perp = 1.45\ \text{Å}^3$ the polarizabilities of $N_2$ along and perpendicular to the molecular axis, respectively[55].

The wavefunction $\Psi$ of an eigenstate is expanded in functions including the angular momentum of $N_2$, the angular momentum of the Ar-axis as well as the intermolecular stretch by the form

$$\Psi = \sum_{j,L} c_{jLJN}(r) <\mathbf{r},\boldsymbol{\Omega}|jLJN> \tag{2}$$

where $j$ and $L$ are the quantum numbers corresponding to $\mathbf{j}^2$ and $\mathbf{L}^2$, respectively. $J$ is the total angular momentum quantum number and $N$ is the quantum number of its projection on the polarization direction of the kick pulse. $\mathbf{r}$ is the vector from the center of mass of the molecule to the atom, $\boldsymbol{\Omega}$ is the vector describing the orientation of the $N_2$ molecule in the space-fixed frame. The expansion parameters are truncated at $J = j = J = 14$. Before applying the alignment pulse, the ground state is obtained by imaginary propagation, and the higher-lying states are obtained by projecting out all the lower-lying states every few imaginary time steps. The initial thermal ensemble is given by the Boltzmann distribution. The alignment degree and intermolecular stretching can be obtained by calculating their expectation values. Here, since the angular distribution is averaged over the angle $\theta$ rather than $\phi$, the isotropic orientation means $<\cos^2\theta> = 1/3$.

## Reporting summary

Further information on research design is available in the Nature Portfolio Reporting Summary linked to this article.

## Data availability

The data that supports the main figures within this study is available from the Zenodo database[56].

## Code availability

The codes that support the findings of this study are available from the corresponding author upon request.

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

## Acknowledgements

The authors thank Anders A. Søndergaard and Henrik Stapelfeldt for sharing the simulation code. We appreciate Hao Liang, Yehiam Prior and Ilya Sh. Averbukh for the fruitful discussions. This work was supported by the National Natural Science Fund (Grants Nos. 12227807, 12241407, 12174109, 92050105), and the Science and Technology Commission of Shanghai Municipality (Grant No. 23JC1402000). R.D. acknowledges support by Deutsche Forschung Gemeinschaft.

## Author contributions

C.L., L.Z., M.S., P.L., K.L., and J.W. contributed to the experiments. C.L., L.X., L.Z., M.S., P.L., W.L., R.D., K.L., and J.W. analyzed the data. All authors contributed to the manuscript.

## Competing interests

The authors declare no competing interests.
