## [Peer Review File · Nature Communications]

Intermolecular interactions probed by rotational dynamics in gas-phase clustersREVIEWER COMMENTS

Reviewer #1 (Remarks to the Author):

Dear Editor,
please find here enclosed my review of the manuscript entitled
"Intermolecular interactions probed by rotational wavepackets in gas-phase clusters", submitted
for publication in Nature Comm.

This work investigates the experimental control of molecular rotation of N_2 molecules subjected to interaction with Ar atoms. They observe that the alignment decays rapidly due to this interaction. They show that the decay rate depends on the relative position of Ar atoms with respect to the molecular axis. Numerical simulations illustrate these experimental observations.

Quantum control is currently a subject of increasing interest in chemistry, physics and mathematics but also for the development of quantum technologies. In this context, the control of molecular rotation is a well-established topic in molecular physics with a variety of applications extending from high harmonic generation to chemical reactivity. This paper focuses on the interaction of aligned molecules with their environment. The proposed method and the results are interesting. To the best of my knowledge, the investigation is original. I am not an experimentalist so I cannot directly judge the relevance of the experimental results. This paper seems sound numerically although I have some doubts about the model system used to reproduce the experimental results.

Here is a series of questions about experiment modeling. The authors describe the quantum system as a pure state. This case corresponds to zero temperature. What is the experimental temperature of the supersonic gas jet? For a non-zero temperature, the system must be described by a density operator. What is the impact of this approximation? The authors consider the interaction of N_2 molecules with one or two Ar atoms. Here again, how to justify this approximation when each N_2 molecule should interact with many atoms. In the literature, the interaction of linear molecules with its environment was described by a Redfield equation (see (1,2) for instance). Can this modeling be used in the case of this paper? This Redfield description also leads to a rapid degradation of molecular alignment. What are the advantages of direct modeling of molecule-atom interaction?

(1)- M. Bournazel et al., *Non-Markovian collisional dynamics probed with laser-aligned molecules*, Phys. Rev. A **107**, 023115 (2023)
(2)- T. Viellard et al., *Field-free molecular alignment for probing collisional relaxation dynamics*, Phys. Rev. A **87**, 023409 (2013)

Reviewer #2 (Remarks to the Author):

The manuscript "Intermolecular interactions probed by rotational wavepackets in gas-phase clusters" by Lu et al explores an internal-rotation wavepacket in N_2 -Ar to probe the dynamics, and potentially coupling, of large-amplitudes internal modes of the molecular system.

Generally, this is a nice experiment that could likely deserves publication in Nat. Comm. following significant clarifications and improvements, mostly regarding the analysis but also with respect to experimental results on the overall rotation of the molecular system.

Firstly, I am wondering if there is no overall-rotation dynamics induced in the N_2 -Ar system by the kick pulse? Obviously, this would be at longer timescales, but should be observable both in the experiment as well as the computations performed by the authors.

Second, I fear in light of >50 years of spectroscopic studies of floppy molecules and molecular clusters, the semantic description does not do justice to the problem. That is, the N₂-Ar system has 3 low-frequency/large amplitude internal "intermolecular" vibrational modes, build up from the lost 3 degrees of translational freedom of the constituents, in addition to the N₂ stretch and the overall rotation of the system. The authors use a cumbersome description of this, in which the N₂-Ar stretch does not seem to occur at all and the two other (internal bending) modes are described as simple, maybe too simple, "perturbed 1D N₂ rotations".

Here, it is clear that these motions are coupled, esp. the various internal degrees of freedom and the overall rotation of the cluster. In fact, in an extremely simplified model the authors derive exactly the first glimpses of that.

Moreover, it seems the overall N₂-Ar stretch of the cluster was "forgotten" in the analysis? Is that justified by any means, i.e., what's the energy gaps and the coupling strengths of this mode to the internal rotations and to the overall rotation?

I also wonder why the degree of alignment even for N₂ itself is so low. Possibly this is due to a relatively large (rotational) temperature (?). What's the explanation of the authors?

In that light and in any case, it is important to understand the "vibrational" temperature for the low-frequency modes of the cluster and in how far the modes are thermally excited before the kick pulse.

Conceptually, when comparing the out-of-plane and the in-plane rotation of the N₂ in the N₂-Ar dimer, how strongly are these modes coupled – to each other, to and via the intermolecular N₂-Ar stretch, and via the overall rotation? Is it appropriate to discuss the results in such one-dimensional models as done in this manuscript?

Are the energies of the relevant (populated) eigenstates in the wavepackets in fact bound states, barrier-modulated above-barrier states, or "free" above-barrier states?

In fact, this question is relevant for both the initially populated states ("temperature") above and the kick-initiated populations.

What's the polarizability anisotropy/tensor of the cluster and how does that change as a function of the intermolecular geometry? On the one hand, this would provide first hints at the overall-rotation dynamics induced by the kick as well as the coupling between internal modes and between internal modes and overall rotation.

When comparing N₂ rotational dynamics to the 1D-internal-rotation dynamics the reduction of dimensionality (1D to 2D) likely needs to be taken into account.

To me, and in light of the very rich set of descriptions the simple model utilized by the authors does not seem to be appropriate. In fact, an analysis of the "intermolecular interactions probed by rotational wavepackets" would be highly appreciated and useful – if actually analyzed and described in the long-standing and proven models of intermolecular interactions. In the simple system presented here, that would require a 6D description (3 degrees of overall rotation and 3 degrees of intermolecular vibrations/internal rotations). This is clearly possible, both semantically as well as computationally.

One of the important points to clarify is which of all these modes are actually directly Raman-excited in the experiment.

Moreover, here it is really important to clarify also experimentally both the intermolecular stretch as well as the overall rotation – and their corresponding interaction with the internal rotations (same for thermal excitations of all these modes).

The authors should also consider to refer to previous alignment experiments of floppy molecules and clusters, which is really what they are following up on here, e.g., PRL 102, (2009); JCP 148, 101103 (2018); PCCP 22, 3245-3253 (2020), ...

In the experimental section, how does a 7:3 splitting of the laser beam lead to intensities that differ by 2 orders of magnitude? Different focusing? Specify. Generally, some more details on the actual experiment should be provided in Methods.

Also, details of the implementation of the computational approach by Zillich should be described.

Re I. 117: No, (N₂)–Ar axis motion can be internal rotation as well as overall rotation...

$|j\rangle$ is a “1D internal rotation” quantum number and should be treated as such.

In the simulations: When the molecule is so floppy, why is it appropriate to use a single reduced mass μ – instead of a coordinate/geometry depended one?

Which pulse intensity and rotational temperature do the simulations yield?

The N₂-Ar₂ cluster seems to be un-analyzed and superfluous. It’s a nice experiment, but it does not provide any information in the current paper. Either leave away or clarify what you learn from this data.

I. 148: “rotational deceleration of N₂ induced by 149 the neighboring Ar in time domain” – is this actually a deceleration or simply a “slower dynamics”? Or is this simply a first dim hint at the coupling of various modes (vide supra)?

Overall, to me this work requires very significant improvements, but I am looking forward to see this then published Nat. Comm.

For clarity, we put the original comments in *italics* to distinguish them from our responses in blue. The text that has been changed or newly added is in red.

REVIEWER COMMENTS

Reviewer #1 (Remarks to the Author):

Dear Editor,

please find here enclosed my review of the manuscript entitled "Intermolecular interactions probed by rotational wavepackets in gas-phase clusters", submitted for publication in Nature Comm.

This work investigates the experimental control of molecular rotation of N₂ molecules subjected to interaction with Ar atoms. They observe that the alignment decays rapidly due to this interaction. They show that the decay rate depends on the relative position of Ar atoms with respect to the molecular axis. Numerical simulations illustrate these experimental observations.

Quantum control is currently a subject of increasing interest in chemistry, physics and mathematics but also for the development of quantum technologies. In this context, the control of molecular rotation is a well-established topic in molecular physics with a variety of applications extending from high harmonic generation to chemical reactivity. This paper focuses on the interaction of aligned molecules with their environment. The proposed method and the results are interesting. To the best of my knowledge, the investigation is original. I am not an experimentalist so I cannot directly judge the relevance of the experimental results. This paper seems sound numerically although I have some doubts about the model system used to reproduce the experimental results.

Here is a series of questions about experiment modeling. The authors describe the quantum system as a pure state. This case corresponds to zero temperature. What is the experimental temperature of the supersonic gas jet? For a non-zero temperature, the system must be described by a density operator. What is the impact of this approximation?

Reply #1:

In the experiment, the rotational temperatures of N₂ and N₂-Ar are estimated to be 11 K and 7 K, respectively.

To make it clear to readers, we have added the following sentences in the Methods:

“The initial temperatures of N₂ and N₂-Ar are estimated to be 11 K and 7 K, respectively, by approximation to their translational temperatures of $T_{trans} = \Delta p^2 / [4 \ln(4) k_B m]$ [see Phys. Rev. Lett. 90, 233003 (2003); J. Chem. Phys. 118, 8699 (2003); Phys. Rev. A 83, 061403(R) (2011)], where k_B is the Boltzmann constant, and Δp and m are the full width

at half maximum (FWHM) of the momentum distribution in the jet direction (Y-axis in our case) and mass of the singly ionized molecules of N_2^+ and $\text{N}_2\text{-Ar}^+$. In our experiment, we measure $\Delta p \sim 3.2$ a.u. and $\Delta p \sim 4.0$ a.u. for N_2^+ and $\text{N}_2\text{-Ar}^+$ ions. The excellent agreement with the experiment is achieved in the simulation when using $T = 9$ K for isolated N_2 and $T = 7$ K for $\text{N}_2\text{-Ar}$ molecule.” (on page 11 line 263)

In our simulation, the initial state is populated according to the measured temperature rather than 0. We also iterate the values used in simulation by comparing the simulation results with the measured alignment trace.

Specifically, we obtain the time-dependent expectation value, $\langle A \rangle_i(t) = \langle \Psi_i(t) | A | \Psi_i(t) \rangle$ for an initial state with the energy of E_i , where $\Psi_i(t)$ is the time-dependent wave function for the i -th initial state. Subsequently, we consider thermal effects by averaging the expectation value over the different initial states with the relative weights given by the Boltzmann distribution,

$$\langle A \rangle(t) = \frac{\sum_i \exp\left(-\frac{E_i}{k_B T}\right) \langle A \rangle_i(t)}{\sum_i \exp\left(-\frac{E_i}{k_B T}\right)},$$

where k_B is the Boltzmann constant and T is the temperature.

Considering the high coupling of the entire system (only the total angular momentum J and its projection N are good quantum numbers in field-free case), we use a collective temperature T to describe the system [J. Chem. Phys. 149, 124301 (2018)]. The excellent agreement with the experiment is achieved when using $T = 9$ K for isolated N_2 and $T = 7$ K for $\text{N}_2\text{-Ar}$ molecule, which are in good agreement with the measurement.

The authors consider the interaction of N_2 molecules with one or two Ar atoms. Here again, how to justify this approximation when each N_2 molecule should interact with many atoms. In the literature, the interaction of linear molecules with its environment was described by a Redfield equation (see (1,2) for instance). Can this modeling be used in the case of this paper? This Redfield description also leads to a rapid degradation of molecular alignment. What are the advantages of direct modeling of molecule-atom interaction?

\(1)- M. Bournazel et al., *Non-Markovian collisional dynamics probed with laser-aligned molecules*, Phys. Rev. A 107, 023115 (2023) \ (2)- T. Viellard et al., *Field-free molecular alignment for probing collisional relaxation dynamics*, Phys. Rev. A 87, 023409 (2013)

Reply #2:

In this work, we focus on the interaction of the N_2 molecule with only one Ar atom both in the experiment and simulation. Although the interaction of the N_2 molecule with two Ar atoms is also measured in the experiment, we cannot simulate such case using our current code. In our work, the molecule-atom interaction is the van der Waals force

formed under low temperature and low pressure, while the molecule-atom interaction in those studies mentioned by the referee is dominated by collisions.

Those studies were performed in a gas cell with a high pressure and the gas density can be up to a few amagat [e.g., 5 amagat corresponding to the density of $1.34 \times 10^{26} \text{ m}^{-3}$ in Phys. Rev. A 107, 023115 (2023)]. In contrast, for the supersonic jet we used, its density was pretty low, typically ranging from 10^{16} to 10^{18} m^{-3} [Phys. Rev. A 68, 023406 (2003); Nat. Phys. 16, 328–333 (2020)], ensuring neglectable collisions with neighboring atoms and molecules. Molecular alignment for typical linear molecules in this case can persist for more than hundreds of picoseconds [Phys. Rev. A 89, 023432 (2014); Phys. Chem. Chem. Phys. 24, 11014 (2022); J. Phys. Chem. A 127, 4848 (2023)].

When collisions dominate as the gas density increases, the phenomena can be described by the Bloch-Redfield form of the Liouville–von Neumann equation [J. Chem. Phys. 124, 034101 (2006)]:

$$\frac{d\rho(t)}{dt} = -\frac{i}{\hbar} [H_0 + H_i(t), \rho(t)] + \left(\frac{d\rho(t)}{dt}\right)_{diss} ,$$

where $\rho(t)$ is the density operator, H_0 is the free rotational Hamiltonian of the rotating molecule, $H_i(t)$ is the molecule-field interaction term.

The collision is exhibited as a dissipative term $\left(\frac{d\rho(t)}{dt}\right)_{diss}$, including the rates of population transfer and elastic collision-induced dephasing. Since it is theoretically impossible to precisely consider numerous atoms/molecules, the collision is only approximated to be time-dependent in the view of the Redfield description. In our case, of only one atom and one molecule, the molecule-atom interaction provides a more comprehensive description as the radical and angular parts of the system are considered.

According to the referee’s comment, we briefly comment the difference in the manuscript:

“Different from the case of high gas densities and temperatures where the collisions dominate the molecule-atom interactions [Phys. Rev. A 107, 023115 (2023), Phys. Rev. A 87, 023409 (2013)], we only consider the interaction of a single N_2 molecule with one Ar atom at low temperatures when the vdW bond is formed.” (on page 7 line 175)

Reviewer #2 (Remarks to the Author):

The manuscript “Intermolecular interactions probed by rotational wavepackets in gas-phase clusters” by Lu et al explores an internal-rotation wavepacket in N₂-Ar to probe the dynamics, and potentially coupling, of large-amplitudes internal modes of the molecular system.

Generally, this is a nice experiment that could likely deserves publication in Nat. Comm. following significant clarifications and improvements, mostly regarding the analysis but also with respect to experimental results on the overall rotation of the molecular system.

Firstly, I am wondering if there is no overall-rotation dynamics induced in the N₂-Ar system by the kick pulse? Obviously, this would be at longer timescales, but should be observable both in the experiment as well as the computations performed by the authors.

Reply #1:

Firstly, we want to clarify that if there was an overall rotation of the N₂-Ar dimer, the first alignment peak of the van der Waals axis should already appear within the current scanning time range (20 ps). Intuitively, the rotation of Ar₂ dimers can be used as a reference because of their comparable rotational constants of 0.05756 cm⁻¹ and 0.07 cm⁻¹ for Ar₂ and N₂-Ar dimers respectively [Phys. Rev. A 83, 061403(R) (2011); Mol. Phys. 27, 903 (1974)].

We have added the following content and a new plot in Fig. 2(c) in the manuscript:

“...On the other hand, Figure 2(c) shows that the vdW axis between the N₂ molecule and the Ar atom, termed as Ar-axis hereafter, hardly responds to the kick. It is worth to mention that our observation window is long enough if the Ar-axis shows an immediate post-pulse alignment. As a reference, the vdW axis of the Ar₂ dimer with a similar rotational constant shows a clear alignment peak at ~ 3.5 ps. The drastically different alignment response for the N-N axis and the Ar-axis inside the N₂-Ar dimer from that for isolated N₂ molecule and Ar₂ dimer indicates that the rotational dynamics can serve as a sensitive probe of the molecule-atom interaction.” (on page 5 line 120)

Figure R1 (Figure 2(c) in the main text). Measured (step horz curves) and calculated

(smooth curves) time-dependent alignment trace of the Ar-axis of Ar₂ and N₂-Ar dimers.

To understand the underlying physics, we performed new simulations by simplifying the N₂-Ar dimer as a rigid asymmetric-top molecule.

“To understand the underlying physics of the rotational dynamics of the N₂ molecule under the influence from the neighboring Ar atom, we performed rigid and floppy model simulations separately. In the rigid model, the N₂-Ar dimer is treated as a rigid asymmetric-top molecule. On the other hand, the vdW bonding nature of the N₂-Ar dimer is implemented by including the molecule-atom interaction potential in the floppy model, where the dynamics of N₂-Ar dimer is described by the internal rotation of N₂ as \mathbf{j} , overall rotation \mathbf{L} and intermolecular stretching (see Methods). Different from the case of high gas densities and temperatures where the collisions dominate the molecule-atom interactions [Phys. Rev. A 87, 023409 (2013), Phys. Rev. A 107, 023115 (2023)], we only consider the interaction of a single N₂ molecule with one Ar atom at low temperatures when the vdW bond is formed.

Figure 2 shows the simulation results for both the rigid and floppy models. Although both results can repeat the decayed alignment traces, the rigid one shows alignment peaks mismatched from the measurement while the floppy model quantitatively agrees with the experiment. This indicates that the decayed alignment trace results from irregular energy spacing induced by the very unique asymmetric-top property of the N₂-Ar molecular structure, i.e. the moment of inertia of the Ar-axis differs drastically from the other two axes, as listed in Table. 1. However, such asymmetric-top property cannot fully account for the observed rotational dynamics. Figures 4(c) and (d) show the rotational spectra for the two models, which clearly illustrates that the rigid model fails to describe the influence of the neighboring Ar atom on the rotational energy of the N-N axis. Intuitively, the rigid model overestimates the red-shift because the Ar-axis is considered to completely corotate with the N-N axis which slows down its rotation in return. Only if the vdW interaction between the N₂ molecule and Ar atom is properly implemented in the floppy model, the rotational spectrum can be reproduced. The comparison between the two models unambiguously reveals that the rotational dynamics of N₂ molecule can serve as a sensitive probe of the neighboring environment. When it comes to the rotation of the Ar-axis, the rigid model considering the total polarizability shows a neglectable alignment response comparing with the Ar₂ dimer. Thus, the vanishing alignment of the Ar-axis stems from the large discrepancy in moments of inertia, i.e., approximately 30 times smaller moment of inertia along the Ar-axis than that along the other axes. The laser pulse induces the alignment of the N-N axis through the fast rotation about the Ar-axis. However, the large moments of inertia of the other two axes lead to the neglectable change in the angular velocities of rotation about the other two axes.” (on page 7 line 169)

Second, I fear in light of >50 years of spectroscopic studies of floppy molecules and molecular clusters, the semantic description does not do justice to the problem. That is, the N₂-Ar system has 3 low-frequency/large amplitude internal “intermolecular” vibrational modes, build up from the lost 3 degrees of translational freedom of the

constituents, in addition to the N_2 stretch and the overall rotation of the system. The authors use a cumbersome description of this, in which the N_2 —Ar stretch does not seem to occur at all and the two other (internal bending) modes are described as simple, maybe too simple, “perturbed 1D N_2 rotations”.

Here, it is clear that these motions are coupled, esp. the various internal degrees of freedom and the overall rotation of the cluster. In fact, in an extremely simplified model the authors derive exactly the first glimpses of that.

Moreover, it seems the overall N_2 -Ar stretch of the cluster was “forgotten” in the analysis? Is that justified by any means, i.e., what’s the energy gaps and the coupling strengths of this mode to the internal rotations and to the overall rotation?

Reply #2:

We thank the referee for the instructive question. The vibration or stretching between the N_2 molecule and the Ar atom is actually observed in our experimental data, as well as included in our simulation. We have added a new Discussion section and a new Fig. 5 to illustrate this issue.

“For isolated diatomic molecules, the large discrepancy between the vibrational and rotational energies ensures decoupling between rotation and vibration. The vibration can be treated individually under an identical radial potential, resulting in orthogonal vibration states. However, this assumption does not hold any more when it comes to the N_2 -Ar dimer since the radial and angular potential is entangled, i.e. the equilibrium distance between the N_2 molecule and Ar atom changes with the relative angle between the Ar-axis and the N-N axis. By assuming the repulsive potential of the two-body fragment channel N_2 -Ar + $n\hbar\omega \rightarrow N_2^+ + Ar^+ + 2e$ as a simple Coulomb potential of $1/r$, the intermolecular distance r as a function of time can be reconstructed. Figure 5(a) shows the expectation value $\langle r \rangle$ as a function of time, which indicates strong vibrational excitation. Although only the polarizability of N_2 is considered in our floppy model, the simulation result well reproduces the intermolecular stretching and its frequency components, as shown in Figs. 5(a) and (b). This indicates that the intermolecular stretching is excited through the coupling with the internal rotation of the N_2 molecule. As marked by the green arrows, states with energy gaps of 15.8 and 20.2 cm^{-1} are assigned to the frequency components ν_1 and ν_2 , respectively. The superposition of these states results in the observed forth-back stretching in Fig. 5(a).

Figure 5(c) displays the transition diagram of the ro-vibrational spectra. As the angular part is expanded onto the coupled basis in the floppy model, approximate quantum numbers (j, J) are employed to label the eigenstates, where j represents the rotational quantum number of the N-N axis (j is not a good quantum number), and only J is the good quantum number of the total angular momentum $J = L + j$. The eigenstates are not assigned to an approximate L -state because of its broad distribution. As numerous states are involved, for illustration, only several states with $J = 4$ are

displayed. For the internal rotation of N₂, states with energy gaps of 9.27, 17.84 and 26.13 cm⁻¹ have been marked by red arrows in Fig. 5(c), which correspond to the observed rotational frequencies in Fig. 4. These frequencies relate to the ones of the N₂ monomer corresponding to the (0–2), (1–3) and (2–4) transitions. Thus, the red-shift of rotational frequencies results from smaller energy gaps for states of the N₂-Ar than the N₂ monomer.” (on page 8 line 204)

Figure R2 (Figure 5 in the main text). Intermolecular stretching of the N₂-Ar dimer. a Reconstructed (blue curve) and calculated (red curve) time-dependent intermolecular stretching of the N₂-Ar dimer. **b** Power spectra of the reconstructed and calculated intermolecular stretching. **c** The energy levels are grouped by the approximate quantum numbers (j, J). The energy gaps related to the frequencies of rotation of N₂ and intermolecular stretching are marked by red and green arrows, respectively.

I also wonder why the degree of alignment even for N₂ itself is so low. Possibly this is due to a relatively large (rotational) temperature(?). What’s the explanation of the authors?

Reply #3:

The reason for the low degree of alignment for N₂ is we use low intensity (7×10^{12} W/cm²) and short duration (50 fs) of the pulse in this work. The temperatures of N₂ and N₂-Ar of the supersonic jet are estimated to be 11 K and 7 K according to their translational temperatures (see Reply #1 of referee1).

In that light and in any case, it is important to understand the “vibrational” temperature for the low-frequency modes of the cluster and in how far the modes are thermally excited before the kick pulse.

Reply #4:

Considering the high coupling of various modes, we use a collective temperature of $T = 7$ K to describe the $\text{N}_2\text{-Ar}$ system [J. Chem. Phys. 149, 124301 (2018)] through which the initial thermal ensemble is given by the Boltzmann distribution $P \sim \exp(-E_j/k_B T)$ with $2J + 1$ degeneracy, where E_j is the energy of an eigenstate state with total angular momentum J . 87% of thermally populated states are summarized in Table I (in the simulation, 99.9% of thermally populated states are involved).

Table I. Thermally populations of eigenstates with $2J + 1$ degeneracy. The states (j, J) are labeled by the quantum numbers of the angular momentum of N_2 , j , and total angular momentum, J .

(j, J)	population	(j, J)	population
0, 0	0.00561	0, 8	0.03482
0, 1	0.01637	1, 8	0.0423
1, 1	0.0198	2, 8	0.0105
0, 2	0.0258	0, 9	0.03023
1, 2	0.03122	1, 9	0.03681
0, 3	0.0332	2, 9	0.00916
1, 3	0.04019	0, 10	0.02528
2, 3	0.00984	1, 10	0.03064
0, 4	0.03816	0, 11	0.01972
1, 4	0.04622	1, 11	0.02465
2, 4	0.01134	0, 12	0.01539
0, 5	0.04055	1, 12	0.01766
1, 5	0.04914		
2, 5	0.01208		
0, 6	0.04051		
1, 6	0.04913		
2, 6	0.01212		
0, 7	0.03843		
1, 7	0.04664		
2, 7	0.01154		

Conceptionally, when comparing the out-of-plane and the in-plane rotation of the N_2 in the N_2 -Ar dimer, how strongly are these modes coupled – to each other, to and via the intermolecular N_2 -Ar stretch, and via the overall rotation? Is it appropriate to discuss the results in such one-dimensional models as done in this manuscript?

Reply #5:

We agree with the referee that the stronger the internal rotation of the N-N axis and the overall rotation are coupled, the more difficult it will be to distinguish between the in- and out of plane rotation of the internal rotation of N-N axis. We have properly illustrated the overall rotation as well as the stretching issue. To make it clearer, we have added the following sentences in the manuscript:

“Benefiting from the statistically super weak alignment response of the Ar-axis, we found that the difference between the in- and out-of -plane rotations survives after selecting the relative angle between Ar-axis and the probe plane.” (on page 6 line 130)

Are the energies of the relevant (populated) eigenstates in the wavepackets in fact bound states, barrier-modulated above-barrier states, or “free” above-barrier states?

In fact, this question is relevant for both the initially populated states (“temperature”) above and the kick-initiated populations.

Reply #6:

The initial thermal population has been answered in Reply #4.

The time-dependent intermolecular distance during and after the laser kick is shown in Fig. R3. If there’s any continuous state or above-barrier state, it will lead to increasing of the molecule-atom distance, i.e. dissociation. As a comparison, the above-barrier state of CH_3I -He is shown. [J. Chem. Phys. 149, 124301 (2018)].

Figure R3. **a** Time evolution of the probability distribution of N_2 -Ar for the distance between the atom and the center of mass of the N_2 molecule. **b** Same as **a** but for CH_3I -He.

What's the polarizability anisotropy/tensor of the cluster and how does that change as a function of the intermolecular geometry? On the one hand, this would provide first hints at the overall-rotation dynamics induced by the kick as well as the coupling between internal modes and between internal modes and overall rotation.

Reply #7:

The polarizability components of the T-shape N₂-Ar are listed in Table II, through which the rigid model is performed. In the floppy model, we only consider the polarizability of the isolated N₂ molecule, the polarizability components along and perpendicular to the molecular axis are set to be $\alpha_{\parallel} = 2.38 \text{ \AA}^3$ and $\alpha_{\perp} = 1.45 \text{ \AA}^3$, respectively.

Strictly speaking, the polarizability changes with the N₂ molecular rotation as well as the stretching. Although we cannot include such complex treatment in our current model, the floppy model still can well reproduce the experimental results, which in return indicates that the dynamical polarizability is not critical here.

We attribute the success to the following reasons. The 50 fs pulse duration in our experiment is much shorter than the period of internal rotation and intermolecular vibration (magnitude of ps), such that the structure deformation during the pulse duration is not important. An improved model including a geometry-dependent polarizability may be required to describe the case of long kick pulse duration as the referee mentioned [e.g., 1.3 ps in Chem. Phys. Lett., 803, 139850 (2022); Phys. Chem. Chem. Phys. 24, 11014 (2022); 150 ps and 1.3 ps in Phys. Chem. Chem. Phys. 22, 3245 (2020)].

Table II. Moments of Inertia and Polarizabilities of the N₂-Ar dimer under T-shape configuration. The *a*-axis and *b*-axis represent the axes parallel to the N-N axis and Ar-axis in the molecular frame, respectively. The *c*-axis is perpendicular to the *a*- and *b*-axes.

Moments of Inertia (amu \AA^2)	Polarizability (\AA^3)
$I_a = 225.54$	$\alpha_{aa} = 3.71$
$I_b = 8.47$	$\alpha_{bb} = 3.32$
$I_c = 234.02$	$\alpha_{cc} = 3.04$

When comparing N₂ rotational dynamics to the 1D-internal-rotation dynamics the reduction of dimensionality (1D to 2D) likely needs to be taken into account.

To me, and in light of the very rich set of descriptions the simple model utilized by the authors does not seem to be appropriate. In fact, an analysis of the “intermolecular interactions probed by rotational wavepackets” would be highly appreciated and useful – if actually analyzed and described in the long-standing and proven models of intermolecular interactions. In the simple system presented here, that would require a 6D description (3 degrees of overall rotation and 3 degrees of intermolecular vibrations/internal rotations). This is clearly possible, both semantically as well as computationally.

Reply #8:

We have explained in Reply #1 and #2. Briefly, we considered internal rotation, overall rotation and intermolecular stretching, while we neglected the vibration between N-N atoms due to its strong interaction

One of the important points to clarify is which of all these modes are actually directly Raman-excited in the experiment.

Reply #9:

Our experimental results indicate that the internal rotation is directly excited through the polarizability of N₂ (Reply #19). For the overall rotation, the vanishing alignment stems from the large discrepancy in moments of inertia (Reply #1). The intermolecular vibration can be excited through coupling with the internal rotation (Reply #2).

Moreover, here it is really important to clarify also experimentally both the intermolecular stretch as well as the overall rotation – and their corresponding interaction with the internal rotations (same for thermal excitations of all these modes).

Reply #10:

We have explained in Reply #1 and #2.

The authors should also consider to refer to previous alignment experiments of floppy molecules and clusters, which is really what they are following up on here, e.g., PRL 102, (2009); JCP 148, 101103 (2018); PCCP 22, 3245-3253 (2020), ...

Reply #11:

We have added the following sentences:

“Recently, investigations on the rotational dynamics of such floppy molecule-atom clusters have been fueled due to their unique properties under the interaction with

neighboring environment [Phys. Rev. Lett. 102, 023001 (2009); J. Chem. Phys. 148, 101103 (2018); Phys. Chem. Chem. Phys. 22, 3245 (2020)]; J. Phys. Chem. A 127, 4848 (2023); Phys. Rev. A 89, 023432 (2014)].” (on page 3, line 59)

In the experimental section, how does a 7:3 splitting of the laser beam lead to intensities that differ by 2 orders of magnitude? Different focusing? Specify. Generally, some more details on the actual experiment should be provided in Methods.

Reply #12:

For clarity, we have added the following sentences in the Method:

“The intensities of them are adjustable using two neutral density filters and a quarter wave plate is placed in the arm of the probe pulse to make it circularly polarized. The probe pulse additionally passed through a telescope which increase its diameter by a factor of 1.5.” (on page 11, line 251)

Also, details of the implementation of the computational approach by Zillich should be described.

Reply #13:

A more detailed description has been added in Method.

Re l. 117: No, (N₂)–Ar axis motion can be internal rotation as well as overall rotation...

Reply #14:

To avoid misleading, we have removed the sentence “For simplicity, we refer to this joint pump probe induced observable as “rotation” ”.

Considering the neglectable alignment of overall rotation (Reply #1), for the rotation dynamics of N₂-Ar, only the internal rotation is a good observable, therefore the rotation refers to the alignment of the N-N axis is caused by internal rotation currently.

/j/ is a “1D internal rotation” quantum number and should be treated as such.

Reply #15:

We have added the description of the floppy model in the revised version. The floppy model considers 3 rotational degrees of freedom and 2 vibrational degrees of freedom, while neglecting the vibration between N-N atoms due to its strong interaction. *j* is the

angular momentum quantum number of the linear molecule.

In the simulations: When the molecule is so floppy, why is it appropriate to use a single reduced mass μ – instead of a coordinate/geometry depended one?

Reply #16:

The reduced mass μ is used to represent the relative motion of Ar and N₂ about their center of mass which is related to the overall rotation and intermolecular stretching. The relative distance (radical part) and rotation (angular part) are responsible for the geometry character of the floppy dimer rather than the reduced mass [J. Chem. Phys. 100, 2505 (1994)].

Thus, the reduced mass doesn't depend on the coordinate or geometry.

Which pulse intensity and rotational temperature do the simulations yield?

Reply #17:

In the simulation, the pulse intensity is set to be 7×10^{12} W/cm² and the initial rotational temperature is set to be 9 K and 7 K for N₂ and N₂-Ar, respectively. See Reply #1 for the first referee.

The N₂-Ar₂ cluster seems to be un-analyzed and superfluous. It's a nice experiment, but it does not provide any information in the current paper. Either leave away or clarify what you learn from this data.

Reply #18:

Under the same laser kick, the internal rotation is more severely hindered by two neighboring Ar atoms. Collaborating with the rotational dynamics of N₂ and N₂-Ar, the absence of the alignment of the N₂ axis in N₂-Ar₂ directly indicates a quantity effect as the number of neighboring atoms increases that what we learned from the experiment. Though a deeper analysis of this complex rotational dynamics is not available currently, this result will be good to know.

l. 148: "rotational deceleration of N₂ induced by 149 the neighboring Ar in time domain" – is this actually a deceleration or simply a "slower dynamics"? Or is this simply a first dim hint at the coupling of various modes (vide supra)?

Reply #19:

For the rotation of N_2 , the central values of each spectral peak, 11.94, 19.9, and 27.86 cm^{-1} , are given by the frequency of a $(j - j + 2)$ coherence and described by the expression $B(4j + 6)$ as a rigid rotor with the rotational constant $B = 1.99 cm^{-1}$.

For the internal rotation of N_2 -Ar, similar to the intermolecular stretching, several states with $J = 4$ have been marked by red arrows in Fig .R2(c). These states with energy gaps of 9.27, 17.84, and 26.13 cm^{-1} are close to the observed rotational frequencies in Fig. 4(b) in the manuscript. Collaborating with approximate labels, these frequencies can relate to the N_2 monomer corresponding to the (0–2), (1–3), and (2–4) coherences.

Thus, the red-shift of rotational frequencies and deceleration result from smaller energy gaps between states of the N_2 -Ar than the N_2 monomer and the quantum beatings of these states induce the slower rotation in the time domain.

As previous studies have suggested, the faster the internal rotation is, the more it behaves like a free rotor, namely a weaker coupling and j becomes a good quantum number. So, the frequency shift directly reflects the coupling strength of the internal rotation with other modes.

Overall, to me this work requires very significant improvements, but I am looking forward to see this then published Nat. Comm.

REVIEWER COMMENTS

Reviewer #1 (Remarks to the Author):

Dear Editor,
since the authors adequately responded to all questions and comments of the referees, I support the publication of this paper.
Best regards,

Reviewer #2 (Remarks to the Author):

Please see attached review.

Reviewer #2 (Attachment):

Review of “Intermolecular interactions probed by rotational dynamics in gas-phase clusters” by Chenxu Lu *et al.*

Overall, the manuscript was strongly improved and all reviewer comments were addressed. Besides a few smaller comments, for clarity and accessibility the description of the molecular dynamics needs to be placed – numerically and semantically – in an the model that takes all relevant dimensions/motions into account. This should likely use the – extremely rich – traditional descriptions of intermolecular interactions in molecular clusters.

I.e., the model of the authors should be correlated to, by explanation, or use traditional models which describe the lost translational degrees of freedom due complex formation as intermolecular vibrations, i.e., stretching, bending, or torsion/internal rotation modes. Textbook descriptions like chapter 9 of Kroto: Molecular Rotation Spectra (1975) or conceptional papers such as *Molecular Physics* **84**, 853-878 (1995) could be a good start for reference. I am convinced the authors can find more such relevant descriptions.

This also includes a – also semantically – combined description of the “internal rotation” and the coupled stretching vibration. Btw. this new data in Fig. 5 is very nice and a strong addition to the paper. Fig. 5c and its description in the initial part of the discussion would strongly benefit from the advanced description sought for in this comment. That the (j, J) nomenclature does not provide a good (supra)molecular frame description of the dynamics is also reflected when it averages the different motions – at least 2 bends and 1 stretch – into one j , which is not even a good quantum number (l. 224).

Generally, a more “supermolecular” description of the cluster might be much clearer to describe the actual couplings and resulting dynamics. This would allow for a direct discussion of the relative contributions of the various mode-to-mode couplings and, furthermore, would allow to put these results into the context of the very many frequency-resolved investigations of such interactions. This would also c

This should also seen in light of the authors insight that “ j is not a good quantum number”

It should be clarified what “properly implemented” in line 192 actually is.

What are “frequency components ν_1 and ν_2 ”?

Some further comments:

When comparing the alignment of the “Ar axis” for N2-Ar and Ar-Ar (Fig 2c), were the corresponding interaction strengths and their anisotropies between the two experiments comparable?

The N2-Ar2 results, as interesting they are, need further description, rationalizing, and discussion to be reasonably included in the manuscript.

Approximating the rotational temperature based on translation can only provide a lower bound for the rotational temperature. In fact, they are often quite different for seeded beams – speed ratios of the beam >100 but still rotational temperatures ~ 10 K. Even if the agreement is good here, this *approximative* approach (“lower bound”) should probably be more clearly pointed out.

Is the indeed relatively low-intensity short-pulse nature of the excitation also the reason for not exciting the overall rotation? Would be worth simulating at higher kick-energy and mentioning in the manuscript.

It would be useful to clearly point out that all dynamics is in truly bound states. Also related to the weak kick strength used.

To me, “deceleration” (reply #19) seems to be the inappropriate word, it seems to say “slower”.

What’s “umbrella time breathing”?

In line 208: Is this truly ‘entanglement’ in “radial and angular potential is entangled”?

Re: NCOMMS-23-43274B

*" Intermolecular interactions probed by rotational dynamics in gas-phase clusters" by
Chenxu Lu et al.*

For clarity, we put the original comments in *italics* to distinguish from our responses in **blue**.
The text that has been changed or newly added is in **red**.

Reviewer #2 (Remarks to the Author):

Overall, the manuscript was strongly improved and all reviewer comments were addressed. Besides a few smaller comments, for clarity and accessibility the description of the molecular dynamics needs to be placed – numerically and semantically – in an the model that takes all relevant dimensions/motions into account. This should likely use the – extremely rich – traditional descriptions of intermolecular interactions in molecular clusters.

I.e., the model of the authors should be correlated to, by explanation, or use traditional models which describe the lost translational degrees of freedom due complex formation as intermolecular vibrations, i.e., stretching, bending, or torsion/internal rotation modes. Textbook descriptions like chapter 9 of Kroto: Molecular Rotation Spectra (1975) or conceptual papers such as Molecular Physics 84, 853-878 (1995) could be a good start for reference. I am convinced the authors can find more such relevant descriptions.

This also includes a – also semantically – combined description of the “internal rotation” and the coupled stretching vibration. Btw. this new data in Fig. 5 is very nice and a strong addition to the paper. Fig. 5c and its description in the initial part of the discussion would strongly benefit from the advanced description sought for in this comment. That the (j, J) nomenclature does not provide a good (supra)molecular frame description of the dynamics is also reflected when it averages the different motions – at least 2 bends and 1 stretch – into one j, which is not even a good quantum number (l. 224).

Generally, a more “supermolecular” description of the cluster might be much clearer to describe the actual couplings and resulting dynamics. This would allow for a direct discussion of the relative contributions of the various mode-to-mode couplings and, furthermore, would allow to put these results into the context of the very many frequency-resolved investigations of such interactions. This should also seen in light of the authors insight that “j is not a good quantum number”

Reply #1:

The floppy model is performed under the space-fixed expression rather than the body-

fixed expression. These two expressions are equivalent, and they are both commonly used descriptions of noncovalent clusters [Chem. Rev., 94, 1931 (1994); Chem. Rev. 100, 4109 (2000)]. We chose the space-fixed (lab-frame) model since it makes us to have a direct comparison with the experimental observations.

Based on the space-fixed expression, a more detailed description of all relevant dimensions (five degrees of freedom) has been added to the Methods

“...The floppy model is conducted under the space-fixed scheme, where the rotation of N₂ and the translation motion are included, and the molecule-atom interaction is represented by an accurate potential surface [J. Chem. Phys. 121, 10419 (2004)]. Due to the complex formation, the angular part and the radical part of the translation motion are reduced to the rotation and intermolecular stretching of the vdW axis, respectively. With the rigid rotor approximation, the vibration of N₂ molecule is assumed to be neglected. Thus, the floppy model includes five degrees of freedom including two rotational degrees of N₂, two rotational degrees and one vibrational degree of vdW axis, and the Hamiltonian in the space-fixed scheme takes the form of...”(on page 12 line 309)

In our floppy model, two rotational degrees of N₂ and two rotational degrees of the vdW-axis are described in the lab frame which is consist with the experimental measurement of their angles with respect to the lab frame Z-axis (polarization direction of the kick pulse) [J. Chem. Phys. 78, 4025 (1983); Chem. Phys. Lett. 221, 161 (1994); Chem. Rev. 94, 1931 (1994); J. Chem. Phys. 108, 3554 (1998)]. For the body-fixed treatment of N₂-Ar, it has been implemented by pioneering works, where the N₂ rotation is described by the angle with respect to the vdW axis in the molecular frame, termed as bending motions [Mol. Phys. 27, 903 (1974); J. Chem. Phys. 88, 578 (1988); J. Chem. Phys. 110, 8525 (1999); J. Chem. Phys. 121, 10419 (2004)].

The main difference of two expressions is the reference frame, and the Coulomb explosion imaging directly images the angular distribution in the lab frame rather than the molecular frame (both for the Ar-axis and N-N axis). Thus, a space-fixed description is more straightforward in our case, especially for the comparison with the isolated N₂ under the same reference frame.

An eigenstate can be expressed as a superposition of different j and L components in the space-fixed scheme (see Methods for the expression of an eigenstate), namely couplings of these modes. Only the parity $p = (-1)^{j+L+J}$, the total angular momentum J and its projection N onto Z-axis are rigorous. To get a qualitatively description about the degrees of freedom involved, approximate quantum numbers are instructive and required for both space-fixed and body-fixed expressions [Chem. Rev. 100, 4109 (2000)]. Collaborating with the approximate quantum number j of N₂-Ar, the frequencies in Fig. 4 can relate to the ones of the N₂ monomer corresponding to the $(j - j + 2)$ transitions, i.e., (0–2), (1–3) and (2–4).

Now approximate quantum numbers (n, j, L) are introduced corresponding to the intermolecular stretching, N₂ rotation and vdW rotation, respectively. Quantum

numbers j and L are assigned according to their dominant components of an eigenstate. The quantum number n is used to represent the intermolecular stretch according to the nodes in the stretch coordinate r [J. Chem. Phys. 88, 578 (1988)].

Besides this qualitative description, a direct knowledge about an eigenstate of its components from relative contributions of various j and L -states, and radical distributions along the stretch coordinate has been added to the Supplementary materials. It can be seen that as the increasing of energy, the quantum number j becomes nearly rigorous and the N_2 rotation behaves like a free rotor, while L still shows a wide distribution attributing to the small rotational energy of Ar-axis.

Figure R1 (Supplementary Figure 1) (a) Populations of j -states of eigenstates (b) Same as (a) but for the L -states. (c) Radical distributions along the stretch coordinate r of eigenstates. The eigenstates (Fig. 5(c) in the manuscript) are assigned by approximate quantum numbers (n, j, L) according to the nodes in the stretch coordinate r , and the most populated components of j and L .

For the intermolecular stretching, we add the following contents to the manuscript:

“...As marked by the green arrows, states with energy gaps of 15.8 and 20.1 cm^{-1} are assigned to the frequency components ν_1 and ν_2 , respectively. The frequency ν_2 originating from the states beating between $n = 0$ and 1 dominates the intermolecular stretching, where n is the approximate stretch quantum number representing the nodes in the stretch coordinate r [J. Chem. Phys. 88, 578 (1988)]. The frequency ν_1 originates from the identical approximate stretch quantum number $n = 0$ of similar but different radial distributions (see Supplementary materials) as n is not rigorous, which results in the stretching with a relatively smaller amplitude as compared to ν_2 .” (on page 9 line 211)

In the revised manuscript, we provide a more detailed description of relevant degrees of freedom of the observables and revise the label in Fig. 5. The pioneer works for N_2 -Ar under the body-fixed expression have been mentioned and cited properly.

“Figure 5(c) displays the transition diagram of the ro-vibrational spectra carried out in the space-fixed expression [Chem. Rev., 94, 1931 (1994)]. Attributing to the low kick intensity which avoids the rotational dissociation [J. Chem. Phys. 147, 074304 (2017); J. Chem. Phys. 149, 124301 (2018)], the observed dynamics occurs in bound states of

N₂-Ar. As the angular part is expanded onto the coupled basis in the floppy model, approximate quantum numbers (n, j, L) are employed to label the eigenstates corresponding to the intermolecular stretching, N₂ rotation and vdW rotation respectively, where j and L are the approximate rotational quantum numbers of N₂ and Ar-axis. An approximate quantum number which is not rigorous implies the involvement of multiple modes within this coordinate. Only the parity $p = (-1)^{j+L+J}$, the total angular momentum J and its projection N are rigorous. As numerous states are involved, for illustration, only several states with even parity and $J = 4$ are displayed. In the view of space-fixed expression, the quantum numbers j and L are not rigorous which means that an eigenstate of N₂-Ar is a superposition of different rotational states of the N-N axis and Ar-axis (see Methods). For simplicity, the approximate quantum number j and L are assigned according to their most populated components (see Supplementary materials). The approximate quantum number is instructive as it builds up a connection between isolated N₂ and N₂-Ar dimer, and reveals behaviors of the N-N axis with different angular momentum under the interaction of Ar. As the increase of j , the N₂ molecule behaves like a free rotor. Besides this space-fixed expression, a molecular frame knowledge about the internal bending of N₂-Ar dimer can be obtained under the body-fixed expression [Mol. Phys. 27, 903 (1974); J. Chem. Phys. 88, 578 (1988); J. Chem. Phys. 110, 8525 (1999)]. For the rotation of N₂, states with energy gaps of 9.27, 17.85 and 26.04 cm⁻¹ have been marked by red arrows in Fig. 5(c), which correspond to the observed rotational frequencies in Fig. 4. Collaborating with the approximate quantum number j of N₂-Ar, these frequencies can relate to the ones of the N₂ monomer corresponding to the (0–2), (1–3) and (2–4) transitions. Thus, the red-shift of rotational frequencies results from smaller energy gaps for states of the N₂-Ar than the N₂ monomer.” (on page 9 line 220)

Figure R2 (Figure 5 in the main text). Intermolecular stretching of the N₂-Ar dimer. **a** Reconstructed (blue curve) and calculated (red curve) time-dependent intermolecular stretching of the N₂-Ar dimer. **b** Power spectra of the reconstructed and calculated intermolecular stretching. **c** The energy levels, with the even parity $p = (-1)^{j+L+J}$ and $J = 4$, are grouped by the approximate quantum numbers (n, j, L) . The energy gaps related to the frequencies of rotation of N₂ and intermolecular stretching are marked by red and green arrows, respectively.

It should be clarified what “properly implemented” in line 192 actually is.

Reply #2:

For clarity, we have modified the corresponding statement :

“Only if the molecule-atom interaction potential rather than a rigid connection between N₂ and Ar is added to describe their interaction (see Methods for details of the floppy model), the rotational spectrum can be reproduced.” (on page 8 line 177)

What are “frequency components ν_1 and ν_2 ”?

Reply #3:

We have explained in Reply #1.

Some further comments:

When comparing the alignment of the “Ar axis” for N₂-Ar and Ar-Ar (Fig 2c), were the corresponding interaction strengths and their anisotropies between the two experiments comparable?

Reply #4:

Both the interaction strengths and anisotropies of them are comparable. The fragments of Coulomb exploded Ar₂ dimer and N₂-Ar dimer are collected in coincidence, ensuring the same kick intensity.

The interaction energy of Ar-Ar has been calculated to be 98.4 cm⁻¹ [J. Chem. Phys. 119, 2102 (2003)] which is close to the one of N₂-Ar (about 100 cm⁻¹) [J. Chem. Phys. 121, 10419 (2004)]. The polarizability anisotropy used in simulation for Ar-Ar is 0.45 Å³ [Phys. Rev. A 89, 023432 (2014)] which is comparable with the one about the Ar-axis in N₂-Ar ($\Delta\alpha_{bc} = \alpha_{bb} - \alpha_{cc} = 0.28 \text{ \AA}^3$).

To make it clearer, we have added the following sentences in the manuscript:

“As a reference, the vdW axis of the Ar₂ dimer with a similar rotational constant, **interaction strength and comparable polarizability anisotropy** [J. Chem. Phys. 119, 2102 (2003); J. Chem. Phys. 121, 10419 (2004); Phys. Rev. A 89, 023432 (2014); Mol. Phys. 27, 903 (1974)] show a clear alignment peak at ~ 3.5 ps” (on page 5 line 117)

The N₂-Ar₂ results, as interesting they are, need further description, rationalizing, and discussion to be reasonably included in the manuscript.

Reply #5:

As we have mentioned in the former reply, from current experiment results, we can obtain a quantity effect as the number of neighboring atoms increases. But currently we cannot simulate such case using our code which prevents us from a further analysis.

To avoid an ambiguous description of N₂-Ar₂, we follow the referee’s former suggestion that we no longer discuss it in the revised manuscript.

Approximating the rotational temperature based on translation can only provide a lower bound for the rotational temperature. In fact, they are often quite different for seeded beams – speed ratios of the beam >100 but still rotational temperatures ~10 K. Even if the agreement is good here, this approximative approach (“lower bound”) should probably be more clearly pointed out.

Reply #6:

We appreciate the reviewer's reminder. To make it clearer, we have added the following sentences in the manuscript:

“The measured translation temperature results in an upper limit of the temperature of molecules in the supersonic gas jet.” (on page 11 line 281)

Is the indeed relatively low-intensity short-pulse nature of the excitation also the reason for not exciting the overall rotation? Would be worth simulating at higher kick-energy and mentioning in the manuscript.

Reply #7:

The alignment signal for the asymmetric-top molecule depends on the molecular properties (polarizability and moments of inertia of three axes), temperature, and the laser parameters. Of course, higher kick intensity leads to higher alignment (see Fig. R3).

However, as we have answered in Reply #4, since the Ar-axes in N₂-Ar and Ar-Ar have similar interaction strengths and their anisotropies are comparable, as well as the kick intensities are the same, it is the large discrepancy of the three molecular axes of N₂-Ar comparing with linear diatomic Ar-Ar that mainly leads to the weak alignment of the Ar-axis rather than the kick intensity.

To make it clear to readers, we have added a corresponding statement

“The kick intensity used here is very low, which might also be the reason that leads to neglectable overall rotation. However, since the Ar-axes in N₂-Ar and Ar-Ar have similar interaction strengths and their anisotropies are comparable, as well as the kick intensities are the same, it is the large discrepancy of the three molecular axes of N₂-Ar comparing with linear diatomic Ar-Ar that mainly leads to the weak alignment of the Ar-axis rather than the kick intensity. On the other hand, when increasing the kick intensity, the rotational dissociation happens [J. Chem. Phys. 147, 074304 (2017); J. Chem. Phys. 149, 124301 (2018)].” page 8 line 189)

Figure R3. Calculated alignment traces of Ar-axis of N₂-Ar and Ar₂ by the rigid model

with low intensity (7×10^{12} W/cm²) and high intensity (1.4×10^{13} W/cm²).

It would be useful to clearly point out that all dynamics is in truly bound states. Also related to the weak kick strength used.

Reply #8:

To make it clearer, we have added the following sentences in the manuscript:

“Attributing to the low kick intensity which avoids the rotational dissociation [J. Chem. Phys. 147, 074304 (2017); J. Chem. Phys. 149, 124301 (2018)], the observed dynamics occurs in bound states of N₂-Ar.” (on page 9 line 221)

To me, “deceleration” (reply #19) seems to be the inappropriate word, it seems to say “slower”

Reply #9:

We have replaced it with a proper description “slower rotation ” in the manuscript.

What’s “umbrella time breathing”?

Reply #10:

The umbrella time breathing means the evolution from alignment to anti-alignment resulting from the beats of the rotational states.

In line 208: Is this truly ‘entanglement’ in “radial and angular potential is entangled”?

Reply #11:

The intermolecular interaction can be described potential energy surface $V(r, \theta_r)$ (see Fig. R3) where θ_r is the angle between the N-N axis and Ar-axis and r is the intermolecular distance between center of mass of the molecule and the atom.

To make it clearer, we have added the following sentences in the manuscript:

“...However, this assumption does not hold any more when it comes to the N₂-Ar dimer since the molecule-atom interaction between N₂ and Ar relies on both their relative distance and angle...” (on page 8 line 200)

Figure R3. Contour plot of potential energy surface for N₂-Ar [J. Chem. Phys. 121, 10419 (2004)]. Geometries are given in \AA and degrees, and energies in cm^{-1} .

REVIEWERS' COMMENTS

Reviewer #2 (Remarks to the Author):

The authors clearly improved the manuscript regarding, and clarified all issues from, the reviews. I suggest to publish the manuscript in Nat. Comm.

I suggest the authors to consider two small comments (without need for review):

1. Regarding "The measured translation temperature results in an upper limit of the temperature of molecules in the supersonic gas jet."

To my mind, the relevant "temperature of molecules" are the rotational and vibrational temperatures of the clusters, which are generally higher than the translational temperature of the beam.

2. It would be useful if the authors or the editorial office could polish the English language of the text, e.g., the latest additions.

Re: NCOMMS-23-43274B

"Intermolecular interactions probed by rotational dynamics in gas-phase clusters"
by Chenxu Lu *et al.*

For clarity, we put the original comments in *italics* to distinguish from our responses in **blue**. The text that has been changed or newly added is in **red**.

Reviewer #2 (Remarks to the Author):

The authors clearly improved the manuscript regarding, and clarified all issues from, the reviews. I suggest to publish the manuscript in Nat. Comm.

I suggest the authors to consider two small comments (without need for review):

1. Regarding "The measured translation temperature results in an upper limit of the temperature of molecules in the supersonic gas jet."

To my mind, the relevant "temperature of molecules" are the rotational and vibrational temperatures of the clusters, which are generally higher than the translational temperature of the beam.

Reply #1:

To make it clear to the readers, we have rephrased the following sentences and corresponding references have been cited properly in the revised manuscript:

“In our experiment, we measure $\Delta p \sim 3.2$ a.u. and $\Delta p \sim 4.0$ a.u. for N_2^+ and $\text{N}_2\text{-Ar}^+$ ions. We expect the measured translation temperature is similar to the rotational and vibrational temperature of molecules in the supersonic gas jet [Phys. Rev. Lett. 90, 233003 (2003); J. Chem. Phys. 118, 8699 (2003); Nat. Phys. 16, 328 (2020)]. The excellent agreement with the experiment can be achieved in the simulation when using the temperatures of 9 K for isolated N_2 and 7 K for $\text{N}_2\text{-Ar}$ molecules.” (line 276 page 11)

2. It would be useful if the authors or the editorial office could polish the English language of the text, e.g., the latest additions.

Reply #2:

We have improved the language of the manuscript.